# MMAR: Towards Lossless Multi-Modal Auto-Regressive Prababilistic Modeling

## Abstract

Recent advancements in multimodal large language models have propelled the development of joint probabilistic models for image understanding and generation. Existing methods that discretize image spaces cause information loss and reduced model capacity. Recent work attempts to integrate diffusion transformers and text autoregression show promise, but it faces challenges in incomplete image information utilization for understanding tasks — diffusion transformers encode image information within various noise levels, but image understanding tasks take only clean image as input. In this paper, we develop a novel MultiModal AutoregRessive (MMAR) probabilistic modeling framework based on continuous image representations. Unlike previous methods, MMAR avoids the information loss associated with discretization and the drawback of combining diffusion transformers with AR models. It employs a standalone diffusion-based continuous probabilistic sampler at the image token level on top of LLMs to theoretically ensure lossless image-text joint probabilistic modeling. In practice, to address the substantial optimization difficulties encountered in low-precision training regime common for LLMs, we theoretically derive an optimal diffusion model parameterization that minimizes numerical error. To balance visual understanding and generalization capabilities, we introduce a two-stage training strategy and an extremely large CFG scale for inference. The proposed MMAR significantly demonstrates scaling-up laws with more data and larger model size. Extensive evaluations are conducted on 18 image understanding benchmarks. It reveals that MMAR is the first joint image-text modeling framework that approaches comparable performance with traditional MLLMs that employ pretrained CLIP vision encoder, marking a significant step toward lossless joint probabilistic modeling of images and text.

## 1 Introduction

Over the past few years, extensive research in the field of multimodal intelligence has catalyzed the accelerated advancement of foundational models for both image understanding and image generation. Within the realm of image understanding, multimodal large language models (MLLM), exemplified by LLaVA (Liu et al., 2023a), have exhibited human-like capabilities in open-domain image comprehension. In the domain of image generation, techniques rooted in generative probabilistic models, such as Denoising Diffusion Probabilistic Models (DDPM) (Ho et al., 2020) and autoregressive (AR) models (Chen et al., 2020), have also garnered significant success. Essentially, these two lines of research correspond to modeling the conditional probability, i.e., $P(T|I)$ and $P(I|T)$, where $T$ and $I$ corresponds to text and image, respectively. It's evident that both types of conditional probabilistic models are subsets of a joint probabilistic model, $P(T, I)$. This brings us to an intriguing question: **Could a joint probabilistic model serve as a natural unifying framework for both image understanding and image generation tasks?**

Given that the most advanced image understanding (Chen et al., 2024) and generation (Esser et al., 2024) models rely on the language priors $p(T)$ of pre-trained large language models (LLMs), the most straight-forward approach for joint image-text probabilistic modeling is to convert images into discrete tokens similar to text. This way, images are treated as a form of language and integrated into the autoregressive modeling of text, as seen in models like MARS, LLAMAGen, and Chameleon. This method leverages the powerful text modeling capabilities of various open-source pre-trained

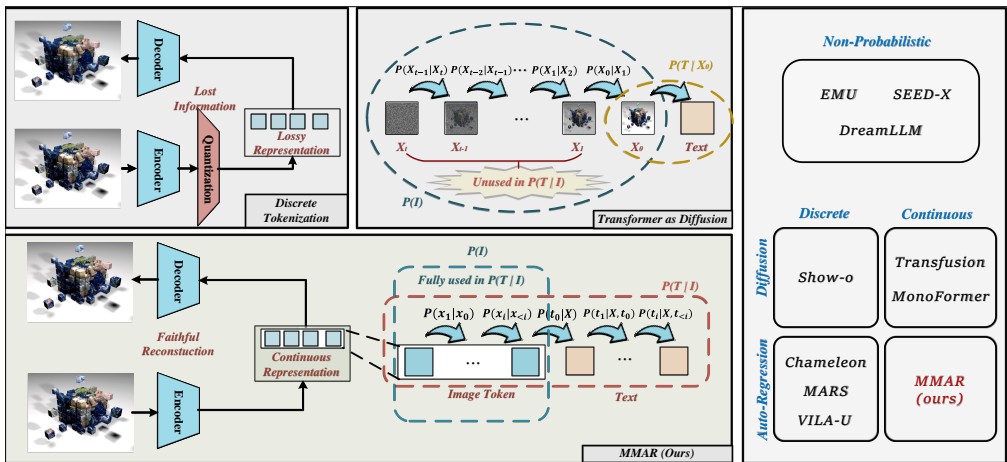

Figure 1: Strengths and weaknesses of different joint image-text probabilistic modeling paradigms

LLMs, along with their highly optimized training and inference frameworks. However, discretization on the continuous image space introduces an information bottleneck, inevitably leading to a loss of image details and reducing the model's information capacity. This limitation is quantitatively evident in the image understanding performance: existing methods based on image token discretization fall short when compared to the LLaVA model, which utilizes continuous CLIP representations (Xie et al., 2024).While there are theoretically feasible methods to mitigate this information bottleneck, such as significantly increasing the number of tokens or the size of the VQ codebook, these approaches would substantially increase the training difficulty for both LLMs and VQ-VAE models.

Recently, several attempts have been made to address the continuous nature of images by combining image diffusion transformers and text autoregression within a unified transformer architecture (Zhou et al., 2024; Zhao et al., 2024). Although this approach, exemplified by Transfusion and MonoFormer (Zhou et al., 2024), demonstrated superior performance compared to discrete token autoregressive methods for images, it is crucial to acknowledge that the inherent differences between diffusion and autoregressive modeling prevents this approach from leveraging complete image modeling capability when it comes to image understanding tasks, as shown in fig. 1. For image diffusion models, they gradually recover a clean image from a low SNR image, and the probability distribution of the image is characterized by the combination of the learned score functions at all noise levels. Existing research (Ho et al., 2020) has illustrated that the information of an image is decomposed and assigned to the score function at different noise levels. For example, at low noise level, the model tends to solve an image enhancement task, while at high noise level, the model is only suitable for extracting rough image morphology and layout. For text autoregressive models, they generate the text tokens from left to right, and the probability distribution is co-depicted by the probability of the next token predicted by each step. In this way, the information of text is assigned to each token of a sentence, and is fully encoded in the hidden states of the model. Ideally, if one wants to ensure the completeness of both image and text information, it is necessary to combine the latent representation of the image at all noise levels together with the entire text token sequence for modeling. However, the typical number of diffusion noise levels usually ranges from tens to one thousand, which makes the training and inference overhead of image understanding unaffordable. From this perspective, transfusion-like methods are trade-offs between information loss and training-inference overhead – they only utilize partial image modeling capability at certain noise levels during training and inference when generating texts from images. As a result, this gap prevents the these approaches from fully utilizing the complete representations learned through image probabilistic modeling for image understanding tasks.

In summary, **how to take in and utilize the complete information of the continuous image modality** is the major pain point of joint image-text probabilistic modeling. This is the challenge that our work is trying to address. The method proposed in this paper, MMAR, belongs to the image-text joint AR methodology, thus naturally avoids the challenge of fusing diffusion and AR, as mentioned previously. Different from other image-text joint AR methods, our method is built upon continuous image representation that can faithfully reconstruct the original image, rather than the discretized lossy image representation. To model the continuous probability distribution within the auto-regressive paradigm, we refer to MAR (Li et al., 2024b), introducing a simple mlp diffusion

sampler to sample continuous image tokens given the transformer backbone output as its conditional information. Different from MAR Li et al. (2024b), we leverage LLMs to learn much more diverse distribution of the large-scale image-text data. However, low-precision training strategies for contemporary LLMs introduce non-negligible numerical error term in the diffusion sampler, which substantially impedes the convergence. We carefully analyze the numerical error term and find that the only way to minimize it is to properly parameterize the diffusion model. By deriving an optimal diffusion parameterization, along with applying two-stage training stragegy and extemely large CFG scale, the lossless multi-modal auto-regressive probabilistic modeling is finally achieved theoretically and practically.

Our contributions are the following three folds:

- We introduce a multi-modal autoregressive probabilistic modeling framework based on continuous image representations, named MMAR. It employs a standalone diffusion-based continuous probabilistic sampler at the image token level on top of LLMs to model continuous image tokens rather than discrete ones, eliminating the information loss typically associated with VQ operations.

- MMAR is designed to model the lossless distribution of large-scale multi-modal data, facing numerous practical challenges. To overcome the key challenge of numerical error, we deriving the optimal diffusion parameterization for low-precision training, and further introduce a two-stage training strategy and extremely large CFG scale to improve the model's image generation and understanding ability.

- Experimental results show that MMAR significantly demonstrates scaling-up laws with more data and larger model size. Extensive evaluations are conducted on 18 image understanding benchmarks, revealing that MMAR is the first joint image-text modeling framework that approaches comparable performance with traditional MLLMs that employ pretrained CLIP vision encoder, marking a significant step toward lossless joint probabilistic modeling of images and text.

## 2 RELATED WORKS

### 2.1 MULTI-MODAL LARGE LANGUAGE MODELS

Since LLMs demonstrated open-domain conversational capabilities, more and more research has be focused on how to introduce visual information into large language models to achieve open-domain visual understanding. Pioneering works such as BLIP-2Li et al. (2023b), MiniGPT4Zhu et al. (2024), LLaVALiu et al. (2023a), etc. use trainable connector modules such as qformer or mlp to align image representations with the input space of LLM, making open-domain visual question answering possible. Recently, thanks to innovations in network structure(Dong et al., 2024b; Tong et al., 2024), training method(Chen et al., 2024) and the support for dynamic resolution input(Hu et al., 2024; Li et al., 2024a), the visual understanding performance of large multi-modal models have been greatly improved. These works focus on the alignment of image representations to text representations, only achieving $p(T|I)$ without incorporating the image's own distribution $p(I)$ into the modeling capabilities of the model. Different from these works, our work adds the modeling of $p(I)$ on the basis of these MLLMs to achieve joint image-text probabilistic modeling.

### 2.2 AUTO-REGRESSIVE IMAGE GENERATIVE MODELS

Text-to-image generative models aim at modeling the conditional probability distribution $p(I|c)$, enabling probabilistic sampling of images conditioned on textual or categorical inputs. Auto-regressive methods represent a dominant paradigm in this domain, typically requiring discrete representations for both input and output. For images, this necessitates encoding them into discrete codes using a VQVAE(Esser et al., 2021; Kondratyuk et al., 2024). While recent works demonstrate that autoregressive methods based on discrete image tokens can generate high-quality images(Sun et al., 2024a), the discretization of image representation acts as an information bottleneck, limiting the modeling accuracy of the image distribution. Recent efforts have shown that autoregressive probabilistic modeling can be achieved without relying on discrete representationsTschannen et al. (2023); Li et al. (2024b). For instance, MAR (Li et al., 2024b) replaces traditional logits with diffu-

sion heads, enabling probabilistic sampling of continuous representations within an autoregressive framework. This paper introduces continuous representation autoregressive probability modeling to MLLMs, mitigating information loss caused by quantization and achieving theoretically lossless joint image-text probability modeling. In addition, we addressed the difficulty when training with large model and large data which is not presented in the previous works.

### 2.3 Unified Visual Understanding and Generation Models

Recently, a series of studies have focused on leveraging a single model to simultaneously address tasks of visual generation and understanding. Early works in this area adopted a modular approach, bridging pre-trained models for visual understanding and visual generation through intermediate representations to achieve combined image understanding and generation. Notable examples include EMU(Sun et al., 2024c;b) and SEED-X(Ge et al., 2024). These works, however, are not considered probabilistic modeling because they aim at modeling the mean value of representations like CLIP or other intermediate representations rather than modeling the true image distribution. This limitation leads to the inadequate image space exploration, and thus hinders the attainment of high-quality generative and understanding performance.

Another line of research adheres to the paradigm of probabilistic modeling (Team, 2024; He et al., 2024; Xie et al., 2024; Wu et al., 2024; Zhou et al., 2024; Zhao et al., 2024). These approaches can be categorized into three types based on whether the image representations are discrete and the modeling method of the image part : (i) Discrete Autoregressive Methods: Examples include Chameleon(Team, 2024), MARS(He et al., 2024), and VILA-U(Wu et al., 2024). These methods discretize image representations and then model images and text tokens using a unified autoregressive transformer. (ii) Discrete Diffusion Methods: An example is Show-o(Xie et al., 2024). These methods discretize image tokens and model them with text tokens using a unified transformer through a discrete diffusion approach. (iii) Continuous Diffusion Methods: Examples include Transfusion(Xie et al., 2024) and MonoFormer(Zhao et al., 2024). These methods do not discretize image representations but directly employ continuous diffusion to model image tokens along with text tokens using a unified transformer. Our approach differs from the aforementioned three types. It belongs to the continuous autoregressive method category, which does not require discretizing image representations. Instead, it predicts the continuous distribution of image tokens using an autoregressive approach and models them alongside text tokens within a unified transformer, as shown in fig.1.

## 3 Method

### 3.1 Auto-Regressive Modeling with Continuous and Discrete Representations

Auto-regressive modeling is a commonly used probabilistic modeling method for sequence data. It can be formulated by "predicting the next token" as follows:

$$\log p_\theta(\mathbf{x}) = \sum_{i=1}^{n} \log p_\theta(x_i|x_{<i}), \tag{1}$$

where $\theta$ and $\mathbf{x}$ represent model parameters and the sequence data, respectively. By maximizing the log likelihood of the data, $\mathbb{E}_{\mathbf{x}\sim\mathcal{D}} \log p_\theta(\mathbf{x})$, the model can be optimized to sample from the data distribution $\mathcal{D}$, achieving probabilistic modeling.

In the realm of natural language processing (NLP), the sequence $\mathbf{x}$ is solely made of discrete text tokens. As a result, most modern large language models (LLMs) parameterize $p_\theta(x_i|x_{<i})$ into a categorical distribution, which can be explicitly represented by the softmax activation on a set of logits predicted by a decoder-only transformer(Radford et al., 2019; Brown et al., 2020) $f_\theta(\cdot)$ with lm head $H_\theta(\cdot)$:

$$p_\theta(x_i|x_{<i}) = \text{softmax}(H_\theta(f_\theta(x_{<i}))). \tag{2}$$

In addition to text, our work also aims at modeling the probability of images, which are represented by continuous rather than discrete image tokens. Therefore, a protocol for parameterizing $p_\theta(x_i|x_{<i})$ of the continuous image tokens is required. Inspired by MAR(Li et al., 2024b), we train a diffusion model to achieve this. The diffusion model takes vector $z_i = f_\theta(x_{<i})$ as the conditional

input, and samples $x_i$ by gradually denoising from a randomly sampled gaussian noise. To optimize the diffusion model for continuous image token sampling, a diffusion loss can be applied, which acts as the upper-bound of the negative log likelihood. A typical diffusion loss can be written as follows, which is seen in MAR (Li et al., 2024b):

$$L(x_i) = \mathbb{E}_{x_i,\epsilon,t}[w_t \cdot ||\epsilon - \epsilon_\theta(\sqrt{\bar{\alpha}_t}x_i + \sqrt{1 - \bar{\alpha}_t}\epsilon, t, z_i)||^2] \geq -\log p_\theta(x_i|x_{<i}) + C, \quad (3)$$

where $w_t$ is the loss weight that balances the loss for different timesteps, and $\bar{\alpha}_t$ indicates the noise schedule of the forward diffusion process. In this way, minimizing diffusion loss is equivalent to maximizing the log likelihood of image data.

The overall loss for joint image-text probabilistic modeling can be written as follows:

$$L = \sum_{i \in I_{img}} L(x_i) - \sum_{i \in I_{txt}} \log p_\theta(x_i|x_{<i}), \quad (4)$$

where $I_{img}$ and $I_{txt}$ indicates the indices of image tokens and text tokens, respectively.

## 3.2 Optimal Diffusion Model Parameterization for Low-Precision Training

In the era of large language models, training with low precision data type, such as `bfloat16`, has become increasingly popular. However, the training and inference process of a diffusion model is relatively sensitive to numerical precision. Moreover, in an auto-regressive framework, the image tokens are sampled sequentially, requiring even more precise sampling for each image token to reduce the overall error accumulation. Therefore, handling the numerical error in the diffusion process modeling should be emphasized when integrating diffusion loss into LLMs.

From the example below, we can clearly illustrate the effect of the numerical error in diffusion models. In eq.3, the diffusion model is parameterized as $\epsilon_\theta(\sqrt{\bar{\alpha}_t}x_i + \sqrt{1 - \bar{\alpha}_t}\epsilon, t, z_i)$, predicting the noise $\epsilon$ that is added to the data. We can explicitly represent the numerical error by multiplying the model prediction by a factor of $(1 + \delta)$, where $\delta$ is the relative error. In this way, we can write DDIM(Song et al., 2021) sampling with numerical error as follows:

$$\tilde{x}^{(t-1)} = \sqrt{\bar{\alpha}_{t-1}} \left( \frac{x^{(t)} - \sqrt{1 - \bar{\alpha}_t}\epsilon_\theta(x^{(t)}, t, z_i)(1 + \delta)}{\sqrt{\bar{\alpha}_t}} \right) + \sqrt{1 - \bar{\alpha}_{t-1}}\epsilon_\theta(x^{(t)}, t, z_i)(1 + \delta). \quad (5)$$

Further separating the numerical error term from the ideal DDIM sampling process, we get:

$$\tilde{x}^{(t-1)} = x^{(t-1)} + (\sqrt{1 - \bar{\alpha}_{t-1}} - \frac{\sqrt{\bar{\alpha}_{t-1}}}{\sqrt{\bar{\alpha}_t}}\sqrt{1 - \bar{\alpha}_t})\epsilon_\theta(x^{(t)}, t, z_i)\delta, \quad (6)$$

where $x^{(t-1)}$ is the ideal DDIM sampling term, and the second term in the above equation is the numerical error term. Assuming $\epsilon_\theta(x^{(t)}, t, z_i)$ follows a standard normal distribution, the standard deviation of numerical error term can be calculated as $|\sqrt{1 - \bar{\alpha}_{t-1}} - \frac{\sqrt{\bar{\alpha}_{t-1}}}{\sqrt{\bar{\alpha}_t}}\sqrt{1 - \bar{\alpha}_t}|\delta$. When the signal-to-noise ratio (SNR) is high, i.e. $\bar{\alpha}_t, \bar{\alpha}_{t-1} \to 1$, the numerical error has almost zero amplitude. However, when SNR is extremely low, i.e. $\bar{\alpha}_t = 0$, and $\bar{\alpha}_{t-1} > 0$, this term will explode to infinity, causing extreme numerical error.

Our goal is to minimize the numerical error term. To achieve this, we analyze all the factors that can determine the numerical error in Appendix A.2, and finally conclude that the only way is to find a proper method to parameterize the diffusion model. By solving the numerical error minimization problem, we conclude that the v-prediction parameterization(Salimans & Ho, 2022) is the desired optimal parameterization method. It is worth noting that "v-prediction" is initially proposed for the efficient distillation of diffusion models, rather than reducing the numerical error of diffusion models. To the best of our knowledge, our work is the first to derive "v-prediction" parameterization from the first principle of minimizing the numerical error in diffusion models, as seen in Appendix A.2. Under v-prediction parameterization, the model predicts a linear combination of data $x_i$ and noise $\epsilon$:

$$v_i^{(t)} = \sqrt{\bar{\alpha}_t}\epsilon - \sqrt{1 - \bar{\alpha}_t}x_i \quad (7)$$

We therefore re-write eq.3 into the v-prediction form, and further set the loss weight $w_t$ to 1 for simplicity:

$$L(x_i) = \mathbb{E}_{x_i,\epsilon,t}[||v_i^{(t)} - v_\theta(\sqrt{\bar{\alpha}_t}x_i + \sqrt{1 - \bar{\alpha}_t}\epsilon, t, z_i)||^2]. \quad (8)$$

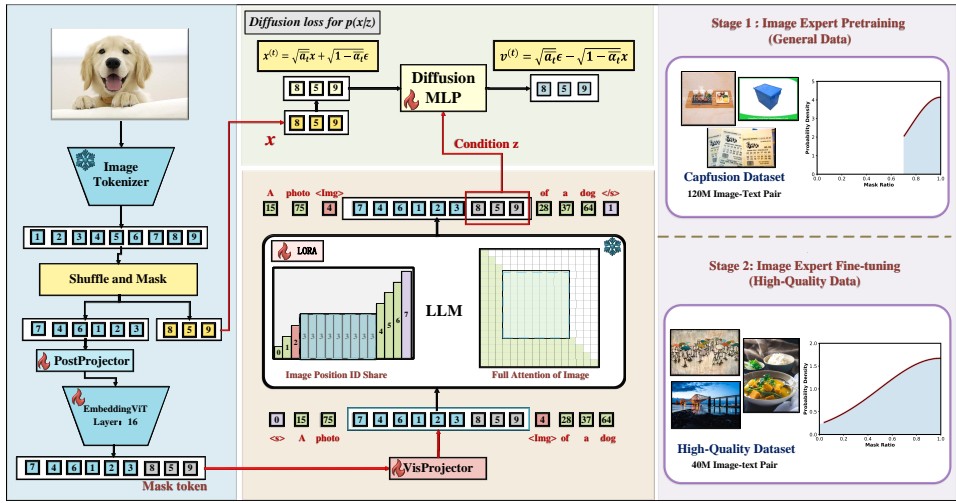

Figure 2: The overview of MMAR with two stage image expert training strategy.

## 3.3 MODEL ARCHITECTURE AND TRAINING STRATEGY

### 3.3.1 PIPELINE

To jointly model the probabilities of images and text at the token level, we employ an image tokenizer to extract image tokens and randomly mask some of them for image generation training, as depicted in fig. 2. Given the substantial distribution gap between images and text, directly scaling image tokens to match the Language Model's (LLM's) dimensions would pose a challenge to joint image-text probability modeling. To address this, we introduce a Latent Encoding process utilizing an EmbeddingViT module for the unmasked image tokens, which simplifies the modeling task. To facilitate the autoregressive training of image tokens, we concatenate the known image tokens with the masked tokens and append the position embedding corresponding to their original positions. These tokens, after being processed by the VisProjector, are concatenated with the text tokens to generate the final image-text sequence. Then, an LLM is used to process the image-text sequence, whose output $z$ integrates information from both the preceding text and the unmasked image tokens. For image output, $z$ acts as a conditional input for the Diffusion MLP, which predicts the $v$ values of the noisy masked image tokens. The diffusion loss, as depicted in eq.8, is applied, enabling us to model the probability of the continuous image tokens. For text output, a linear lm-head processes $z$ to predict the next token's logits, and the corresponding cross entropy loss is applied, allowing us to model the probability of the text tokens.

During image generation, to ensure consistency with the training scenario, we initialize all image tokens as mask tokens and randomly generate a set of image position sequences. Following a left-to-right order, we extract N condition tokens $z$ at each step and input them into the Diffusion MLP to generate the corresponding image tokens. Next, we assign these image tokens back to their respective positions in the original image token sequence, iterating until the entire image token set is obtained. Ultimately, we decode the image tokens into images using the Image Tokenizer, yielding the generated images.

### 3.3.2 MODEL ARCHITECTURE

**Tokenizer** We employed publicly available tokenizers provided by LDM (Rombach et al., 2022) for our experiments. We implemented two versions, VQ-16 and KL-16. It is important to note that no updates were made to the tokenizers during the course of our training.

**EmbeddingVit** To achieve a more profound encoding of image tokens, we constructed a EmbeddingViT with 16 vit block layers(Dosovitskiy et al., 2021), featuring 1024 hidden state channels. Moreover, as the input consists of randomly shuffled image tokens, we integrated a learnable position embedding for each image token in EmbeddingViT, corresponding to its original position.

**LLM** We initialized our LLM using parameters from the open-source QWen2 series models (Yang et al., 2024). To preserve text modeling capabilities, we kept the LLM parameters fixed during train-

ing and added independent Lora layers specifically for image token training. Specifically, we added two additional linear layers for each linear layer in the original model, and only the image tokens in the input pass through these two linear layers. We refer to this LoRA approach as Plora (a type of expert). Considering the time cost, our ablation experiment employed QWen2-0.5B-Instruct, with Plora having 512 intermediate layer channels. Meanwhile, we used QWen2-7B-Instruct to explore our method's scale-up capability and performance ceiling, with Plora having 1280 intermediate layer channels.

In the LLM, we adopted a bidirectional attention mechanism to enhance information exchange between image tokens, rendering all image tokens mutually visible, as depicted in fig. 2. Meanwhile, the text portion retains causal attention. Additionally, to prevent interference with the autoregressive training of image tokens in random order, we assigned the same position to all position IDs in the image part before calculating ROPE within the LLM. This strategy not only ensures the random order regression of image tokens but also mitigates the issue of text concentrating on closer image tokens in long text scenarios.

**Diffusion MLP**   Drawing inspiration from MAR, we also employed a simple MLP to predict the $V$ value. This MLP consists of a series of residual blocks, each comprising AdaLN (Peebles & Xie, 2023), a linear layer, SiLU, and another linear layer. The conditional representation $Z$ is added to the corresponding time embedding and incorporated through AdaLN. In the QWen2-0.5B-Instruct experiments, we configured the Diffusion MLP to have 8 layers of blocks and a width of 1024 channels. Conversely, in the QWen2-7B-Instruct experiments, we set the Diffusion MLP to have 12 layers of blocks and a width of 2048 channels.

### 3.3.3   TRAINING STRATEGY

**Multi Task**   To accomplish joint image-text modeling, we simultaneously conducted text-to-image, image-to-text, and unconditional image generation tasks during training. In the image-to-text task, no mask tokens are assigned to the image part, allowing us to model $P(T|I)$ using the complete image tokens. Furthermore, the allocation ratio of text-to-image and unconditional image generation tasks is set to 9:1, facilitating efficient use of the cfg technique at the inference stage. To maintain a balance between tasks, we intuitively set the sample allocation ratio of image generation tasks and image understanding tasks to 1:1.

**Two Stage Training**   To achieve a balance between image generation and understanding capabilities, our training process is carried out in two stages. The first stage utilizes large-scale, mid-quality data (as illustrated in fig. 2) to enhance the diversity of the model's data distribution (Hu et al., 2024). By initially modeling this diverse data, we strengthen the model's understanding capabilities. In the second stage, we employ a smaller volume of high-quality data to further improve the image generation capacity and refine the model's comprehension of images.

We observe that when the training dataset is excessively large, preventing the model from iterating over each image hundreds of times, maintaining an image mask ratio within the range of [0.7, 1], as done in MAR, hinders the model's ability to generate coherent image tokens. Consequently, in the second stage, we adjusted the mask ratio to (0, 1]. The example is showed in Appendix A.5. The probability density curves of the two mask ratios are shown on the right side of fig. 2. Additionally, due to the small size of the diffusion mlp and the limited amount of high-quality data, we increased the number of timestep samples for the given $z$ to enhance learning efficiency. To further mitigate performance fluctuations in the model, we employed an exponential moving average (EMA) method with a momentum of 0.9999 in both stages.

### 3.3.4   INFERENCE STRATEGY

**Classifier-Free Guidance**   During the inference stage, the model performs both text-based and non-text-based image generation tasks. The provided conditions are represented as $z_c$ and $z_u$, and the predicted $v$ is given by: $v = v_\theta(x^{(t)}|t, z_u) + \omega * (v_\theta(x^{(t)}|t, z_c) - v_\theta(x^{(t)}|t, z_u))$ (which has the same effect as $\epsilon = \epsilon_u + \omega * (\epsilon_c - \epsilon_u)$, with the mathematical derivation process detailed in the Appendix A.4), where $\omega$ denotes the guidance scale. Our method has been experimentally validated to support a large guidance scale, as compared to the noise prediction training approach. We hypothesize that this is due to the strong conditioning provided by $z$, which, coupled with the

Table 1: Comparison on visual understanding benchmarks. MMAR surpasses other joint image-text probabilistic models by a large margin, even with a small resolution of 256x256, approaching the performance of traditional MLLMs like LLaVA, which employ pretrained CLIP vision encoder.

| Method | LLM | V-Token | Res. | MMB | MME$^P$ | POPE | SEED | MM-Vet | AVE@18Und. |
|---|---|---|---|---|---|---|---|---|---|
| LLaVA-1.5 | Vicuna-1.5-7B | CLIP | 336 | 64.3 | **1510.7** | **85.9** | 58.6 | 31.1 | **47.08** |
| EMU-2 | LLaMA-13B | CLIP | 448 | – | – | – | 62.8 | **48.5** | – |
| SEED-X | LLaMA-13B | CLIP | dynamic | **75.4** | 1435.7 | 84.2 | – | – | – |
| DreamLLM | Vicuna-7B | CLIP | 224 | 58.2 | – | – | – | 36.6 | – |
| Chameleon-7B | 7B from scratch | vq-vae | 512 | 13.32 | 125.8 | 30.86 | 34.61 | 7.34 | 18.34 |
| Transfusion* | Qwen-2-0.5B | vae | 256 | 29.47 | 594.3 | 66.90 | 42.40 | 13.90 | 28.26 |
| Show-o | Phi-1.5B | CLIP | 336 | 42.44 | 1182.7 | 84.50 | 51.61 | 20.87 | 33.06 |
| VILA-U | LLaMA-2-7B | vq-vae | 256 | – | 1336.2 | 83.9 | 56.3 | 27.7 | – |
| MMAR-0.5B | Qwen-2-0.5B | vae | 256 | 48.45 | 882.1 | 70.74 | 55.70 | 18.49 | 34.56 |
| MMAR-7B | Qwen-2-7B | vae | 256 | 66.32 | 1393.9 | 83.02 | **64.52** | 27.80 | 46.52 |

autoregressive (AR) process, necessitates reducing accumulated error. Thus, each token must be generated as accurately as possible, a requirement that can be fulfilled with a large guidance scale.

# 4 EXPERIMENT

## 4.1 DATASET

We utilized the Capfusion-120M dataset for the image expert pretraining stage. This dataset is publicly accessible and comprises an extensive collection of web-based image-text pairs, designed to optimize noisy captions (Yu et al., 2023). In an effort to further improve the quality of the content generated, we executed a random sampling of 20M data points from the Capfusion dataset during our image expert fine-tuning stage. This was supplemented with a high-quality mixed dataset that included ImageNet-1K-1.2M, CC12M, and laion-aesthetics-12m. Importantly, we employed the open-source InternVL2-8B for recaptioning the CC12M and laion-aesthetics-12m datasets in English. Following LLaVA-v1.5 (Liu et al., 2023a), we use LLaVA-v1.5-mix-665K for instruction tuning before each performance test for understanding.

## 4.2 IMPLEMENTATION DETAILS

By default, we utilized the AdamW optimizer with betas (0.9, 0.95). The weight decay was consistently maintained in proportion to the learning rate. During the first training stage, a learning rate of 5e-5 was employed for a total of 4 epochs, with an initial warm-up phase of 0.5 epochs, followed by maintaining the learning rate at 5e-5. In the second stage, the 0.5B model maintained a 5e-5 learning rate and a 0.5 epoch warm-up, with training lasting for 3 epochs. For the larger 7B model, we initially applied a 2e-6 learning rate for 1.5 epochs, before transitioning to a 1e-6 learning rate for an additional 1.5 epochs. Furthermore, during the first stage, the total batch size for the smaller 0.5B model was 2496, while it was 1152 for the larger 7B model. In the second stage, the total batch size for the smaller 0.5B model was 768, and for the larger 7B model, it was 480. Notably, in both stages, only the Lora portion of the LLM parameters was released, and a consistent image resolution of 256x256 was used throughout. Moreover, our models exclusively utilized the `bfloat16` data type during training, while `float32` was applied in the DDIM or DDPM sampler during inference.

## 4.3 COMPARISON WITH OTHER SYSTEMS

**Visual Understanding.** As depicted in table 1, we thoroughly gauge MMAR's performance in visual understanding, employing VLMEvalKit(Duan et al., 2024) to perform extensive evaluations on prevalent visual understanding benchmarks, encompassing a total of 18 such assessments (average score denoted by "AVE@18Und." in table 1) including MMB(Liu et al., 2023b), MME(Fu et al., 2023), POPE(Li et al., 2023c), SEED(Li et al., 2023a), MM-Vet(Yu et al., 2024), among others. Our method outperforms other joint image-text probabilistic models by a large margin, including Chameleon-7B(Team, 2024), Show-oXie et al. (2024), VILA-UWu et al. (2024) and our re-implemented version of Transfusion (denoted by "Transfusion*"), approaching the performance of traditional MLLMs like LLaVA, which employ pretrained CLIP vision encoder. Even without using any pre-trained CLIP or diffusion models and with small resolution of $256 \times 256$, MMAR-7B presents comparable or even better performance when compared to methods using pre-trained clip

and diffusion models, including EMU-2(Sun et al., 2024c), SEED-X(Ge et al., 2024), and Dream-LLM(Dong et al., 2024a).

**Visual Generation.** We showcase the zero-shot FID of MMAR, evaluated on the MSCOCO 30k dataset, in Table 2. Our model's performance is discernibly on par with current robust generative models, a noteworthy achievement for MMAR given its minimal training epochs. Repeated exposure to identical data can substantially enhance a model's generative quality, as exemplified by MAR, which trains for 400 epochs on the 1.2M ImageNet dataset, and Show-o, which undergoes approximately 49 epochs of training on the

Table 2: Comparison on MSCOCO Dataset

| Type | Method | Params | Images | FID-30K↓ |
|------|--------|--------|--------|----------|
| Gen. Only | DALL-E | 12B | 250M | 27.50 |
|  | LDM | 1.4B | 400M | 12.64 |
|  | DALL-E2 | 6.5B | 650M | 10.39 |
|  | Imagegen | 3B | 5000M+ | 6.61 |
| Und. and Gen. w/ pre-trained Diff. | CoDI | - | 400M | 11.26 |
|  | SEED-X | 17B | - | 12.68 |
|  | DreamLLM | 7B | - | 8.76 |
| Joint Prob. Models | Show-o | 1.3B | 35M | 9.24 |
|  | Chanmeleon | 7B | - | 29.6 |
|  | Transfusion[1] | 7B | - | 16.8 |
|  | MMAR-0.5B | 0.5B | 145.2M | 36.6 |
|  | MMAR-7B | 7B | 145.2M | 17.1 |

35M dataset. However, to bolster the model's comprehension capabilities, exposure to diverse data is crucial rather than simply reiterating the same data. Consequently, we restrict our training to just 3 epochs for the second stage high-quality dataset. Despite these constraints, our model maintains competitive performance in generation, further substantiating the efficacy of our image-text joint modeling approach. The example images generated by our model are shown in the Appendix A.6.

**Scaling up with Model Size.** From Table 1 and 2, we see that MMAR can scale up from 0.5B to at least 7B model size, with significant improvement of visual understanding and generation capability.

## 4.4 ABLATION STUDY

We begin by evaluating the impact of our chosen diffusion parameterization method. Table 3 demonstrates that switching to the more common n-prediction leads to a significant decrease in both visual understanding and generation quality, confirming the effectiveness of our optimal diffusion parameterization. To evaluate the efficacy of various image-text joint modeling techniques, we devised two distinct versions based on the MMAR framework: one employing VQ-16 for discrete token modeling, and the other utilizing Transformer for diffusion modeling (refer to Transfusion). Comprehensive implementation details are provided in the Appendix A.1. Our test results are presented in Table 3. The Transformer-based diffusion modeling version considerably underperforms the other two approaches in both understanding and generation aspects. This is attributed to the substantial loss of image information when jointly modeling image-text and operating with limited training epochs. In contrast, our method consistently delivers superior results.

Table 3: Ablation study on MMAR

| Type | MMB | $MME^P$ | POPE | SEED | MM-Vet | AVE@18Und. | FID-30K↓ |
|------|-----|---------|------|------|--------|------------|----------|
| MMAR-0.5B(Full method) | **48.45** | **882.1** | **70.74** | **55.70** | **18.49** | **34.56** | **36.6** |
| w/ **n**-pred. | 45.53 | 880.7 | 71.14 | 53.72 | 17.98 | 32.21 | 61.53 |
| w/ VQ | 37.54 | 618.2 | 66.98 | 44.93 | 14.45 | 29.70 | 66.26 |
| transfusion-like | 29.47 | 594.3 | 66.90 | 42.40 | 13.90 | 28.26 | 95.38 |

## 4.5 ANALYSIS

**Impact of v-prediction** To delve deeper into the disparities between v-prediction and n-prediction for diffusion MLP, we independently collect the statistics of the MSE Loss of $v^{(t)}$ values at various time steps $t$ throughout the training process for both methods, as illustrated in fig. 3 (A). Furthermore, to more effectively discern the loss discrepancies between the two techniques, we subtracted the n-prediction curve from the v-prediction curve, yielding the yellow curve. The red line is the theoretical numerical error of n-prediction, as discussed in Appendix A.3. The graph reveals that the loss of n-prediction model is consistently higher than v-prediction model, especially when $t > 900$, curve $n - v$ exhibits a significant spike towards infinity. This aligns with the behavior of the theoretical numerical error, confirming the non-negligible numerical error effect in low-precision training.

---

[1]Both the Transfusion and Chanmeleon results are referenced from Table 3 in the paper 'Transfusion: Predict the Next Token and Diffuse Images with One Multi-Modal Model.'

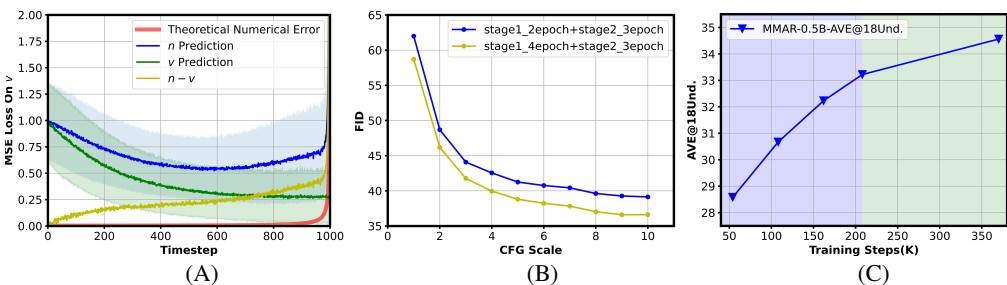

Figure 3: The impact of v-prediction, CFG scaling, and training steps.

The gap between yellow and red curve indicates that apart from the direct effect, numerical error also introduces optimization difficulty, hindering the loss convergence.

**Impact of CFG Scaling** We selected models from the second and fourth epochs of the first stage as starting points for the second stage, trained them for 3 epochs, and then tested the MSCOCO FID-30K under varying CFG intensities. As shown in fig. 3 (B), our method achieves better FID scores as the CFG scale increases from 1 to 10. It is worth noting that most probabilistic generative models typically have a CFG scale between 1.5 and 5. Additionally, a longer training duration in the first stage (4 epochs) results in better generation outcomes, confirming our decision to train for 4 epochs initially. This decision was driven by the need to ensure both performance understanding and improved generation quality.

**Scaling up with More Training Data** As illustrated in fig. 3 (C), we evaluate the average performance of MMAR-0.5B on 18 comprehension task benchmarks, following SFT training applied to the checkpoints generated during its training process. The blue background in the figure denotes the Image Expert Pretraining stage, while the green background signifies the Image Expert Fine-tuning stage. The curve reveals that, with an increasing number of training steps, i.e. more training data, the comprehension performance of MMAR-0.5B consistently improves without reaching saturation. This finding highlights the exceptional scale-up capability of our MMAR model.

### 4.6 Limitation

Our method still requires further optimization in terms of image generation speed. Although the diffusion mlp is not involved in image understanding tasks, when applied to image generation tasks, we are compelled to use a 256-step token generation followed by a 100-step diffusion denoising process to ensure the quality of the generated images. This results in a generation time of nearly three minutes for a single 256x256 image. While it is possible to generate multiple images simultaneously by increasing the batch size, this does not fundamentally resolve the issue of prolonged generation times. We plan to address this in future work.

## 5 Conclusion

This paper proposes MMAR, a novel multimodal auto-regressive probabilistic modeling framework based on continuous image representations. It employs a standalone diffusion mlp at the image token level on top of pre-trained LLMs to facilitate theoretical lossless joint image-text probabilistic modeling. In practice, the low precision training of LLMs poses an non-negligible numerical error term to diffusion loss, causing optimization difficulty. This was addressed by deriving an optimal diffusion model parameterization. To balance the understanding and generation ability, a two-stage training strategy is introduced. During inference time, MMAR can tolerant extremely large CFG scale to generate high quality images. MMAR significantly demonstrates scaling-up laws with more data and larger model size. Extensive evaluations are conducted on 18 image understanding benchmarks, revealing that MMAR is the first joint image-text modeling framework that approaches comparable performance with traditional MLLMs that employ pretrained CLIP vision encoder, marking a significant step toward lossless joint probabilistic modeling of images and text.

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

# A APPENDIX

## A.1 ADDITIONAL IMPLEMENTATION DETAILS

**VQ**   Based on the MMAR-0.5B framework, we replace the Image Tokenizer from KL-16 to VQ-16. The image codes extracted using VQ-16 are then passed through a projector to increase the channel size to match the LLM's hidden size. Subsequently, we add a decoding Linear layer, which takes the hidden states of the LLM's output image portion as input and maps them to the image codebook. The Cross Entropy loss is then calculated between these mapped values and the actual VQ codes.

**Transfusion**   Following the theoretical ideas presented in the Transfusion paper, we adopt a simple linear mapping. After extracting image tokens using KL-16, if the task is image first, we add noise within a 500-time step to the image tokens. Otherwise, we add noise within a 1000-step. After the linear mapping, we add the image token a learnable time embedding corresponding to the time step as input to the LLM. We also maintain the bidirectional attention mechanism.

After passing through the LLM, we first map the output back to the original token channel count using a linear layer and compute the MSE loss for the predicted noise. During generation, we treat LLM as a denoised model, with the condition being the concatenation of the text and the image tokens to be generated. We adopt the learning approach of Transfusion but conduct experiments based on our training tasks and stage divisions.

**Projector**   In order to accomplish channel alignment, we introduced two Projectors designed to scale the channels. Both Projectors consist of a simple linear layer and multiple blocks composed of activation layers and linear layers. The PostProjector comprises one block, whereas the VisProjector contains two blocks.

## A.2 MINIMIZING THE NUMERICAL ERROR IN DIFFUSION MODELS

To make our discussion clearer, we switch the diffusion noise schedule into an angular form as follows:

$$\begin{cases} \sin \phi_t = \sqrt{1 - \bar{\alpha}_t}, \\ \cos \phi_t = \sqrt{\bar{\alpha}_t}. \end{cases} \tag{9}$$

In this way, the forward diffusion process can be written as follows:

$$x^{(t)} = \sqrt{\bar{\alpha}_t} x + \sqrt{1 - \bar{\alpha}_t} \epsilon = \cos \phi_t x + \sin \phi_t \epsilon, \tag{10}$$

where $x^{(t)}$, $x$ and $\epsilon$ are noised image latent, original image latent and gaussian noise, respectively.

Our goal is to minimize the numerical error term in the DDIM sampling process. However, the form of DDIM sampling process is different under different parameterization method of the diffusion model. Therefore, we need to first define a general form to represent the diffusion model parameterization.

We consider the diffusion model output $u_\theta^{(t)}$ predict a linear combination of data $x$ and noise $\epsilon$, i.e. $u^{(t)} = a_t x + b_t \epsilon$. Note that the coefficients can vary according to the diffusion time step $t$. Further re-writing the coefficients in the angular form gives:

$$u^{(t)} = r_t \cos \psi_t x + r_t \sin \psi_t \epsilon, \tag{11}$$

where $r_t = \sqrt{a_t^2 + b_t^2}$ represents the scale of $u^{(t)}$. $\cos \psi_t$ and $\sin \psi_t$ balance the proportion of $x$ and $\epsilon$. Combining eq.10 and eq.11, we can in turn represent $x$ and $\epsilon$ with $u^{(t)}$ and $x^{(t)}$:

$$\begin{cases} x = \frac{\sin \psi_t x^{(t)} - \sin \phi_t u^{(t)}/r_t}{\cos \phi_t \sin \psi_t - \cos \psi_t \sin \phi_t} = \frac{\sin \psi_t x^{(t)} - \sin \phi_t u^{(t)}/r_t}{\sin(\psi_t - \phi_t)}, \\ \epsilon = \frac{\cos \psi_t x^{(t)} - \cos \phi_t u^{(t)}/r_t}{\sin \phi_t \cos \psi_t - \sin \psi_t \cos \phi_t} = -\frac{\cos \psi_t x^{(t)} - \cos \phi_t u^{(t)}/r_t}{\sin(\psi_t - \phi_t)}. \end{cases} \tag{12}$$

Now we consider the general form of DDIM sampling stepSong et al. (2021):

$$x^{(t-1)} = \cos \phi_{t-1} \hat{x}_\theta(x^{(t)}) + \sin \phi_{t-1} \hat{\epsilon}_\theta(x^{(t)}), \tag{13}$$

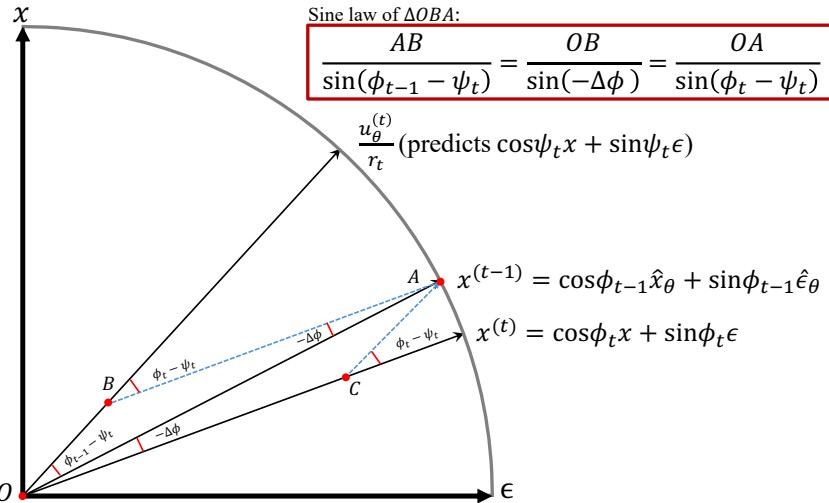

Figure 4: Geometric interpretation of a DDIM sampling step under arbitrary diffusion model parameterization.

where $\hat{x}_\theta(x^{(t)})$ and $\hat{\epsilon}_\theta(x^{(t)})$ are the estimated image latent and noise, respectively.

Note that by using eq.12, both of $\hat{x}_\theta(x^{(t)})$ and $\hat{\epsilon}_\theta(x^{(t)})$ can be derived from the noisy image latent $x^{(t)}$ and the diffusion model output $u_\theta^{(t)}$. Therefore, we can further represent $x^{(t-1)}$ in the following form:

$$x^{(t-1)} = \cos\phi_{t-1} \frac{\sin\psi_t x^{(t)} - \sin\phi_t u_\theta^{(t)}/r_t}{\sin(\psi_t - \phi_t)} - \sin\phi_{t-1} \frac{\cos\psi_t x^{(t)} - \cos\phi_t u_\theta^{(t)}/r_t}{\sin(\psi_t - \phi_t)}$$

$$= \frac{\sin(\phi_{t-1} - \phi_t)u_\theta^{(t)}/r_t - \sin(\phi_{t-1} - \psi_t)x^{(t)}}{\sin(\psi_t - \phi_t)}. \tag{14}$$

Eq.14 represents the general form of DDIM sampling step under any kind of diffusion model parameterization in the form of eq.11. To help understanding, we further present the geometric meaning of eq.14. As shown in fig.4, term $x^{(t-1)}, x^{(t)}$, and $u_\theta^{(t)}/r_t$ all locate on the unit circle in the $x - \epsilon$ plain. We find that eq.14 can be interpreted as projecting $x^{(t-1)}$ onto the $(x^{(t)}, \frac{u_\theta^{(t)}}{r_t})$ coordinate system. We illustrate this projection by adding auxiliary line $AB$ and $AC$. By solving the sine law of $\triangle OBA$ given $OA = 1$, we get:

$$\begin{cases} OB = \frac{\sin(\Delta\phi)}{\sin(\psi_t - \phi_t)} \\ BA = \frac{-\sin(\phi_{t-1} - \psi_t)}{\sin(\psi_t - \phi_t)} \end{cases} \tag{15}$$

By representing $x^{(t-1)} = OB \cdot u_\theta^{(t)}/r_t + AB \cdot x^{(t)}$, we get:

$$x^{(t-1)} = \frac{\sin(\Delta\phi)}{\sin(\psi_t - \phi_t)} u_\theta^{(t)}/r_t - \frac{\sin(\phi_{t-1} - \psi_t)}{\sin(\psi_t - \phi_t)} x^{(t)}, \tag{16}$$

which aligns with eq.14 given that $\Delta\phi = \phi_{t-1} - \phi_t$.

Now, we take the numerical error into consideration by multiplying the model output by a factor $1 + \delta$, where $\delta$ represents the relative error:

$$\tilde{x}^{(t-1)} = \frac{\sin(\phi_{t-1} - \phi_t)(1 + \delta)u_\theta^{(t)}/r_t - \sin(\phi_{t-1} - \psi_t)x^{(t)}}{\sin(\psi_t - \phi_t)}. \tag{17}$$

Further, we can isolate the numerical error term from the ideal DDIM sampling step:

$$\tilde{x}^{(t-1)} = x^{(t-1)} + \sin(\phi_{t-1} - \phi_t)\frac{u_\theta^{(t)}/r_t}{\sin(\psi_t - \phi_t)}\delta. \tag{18}$$

From eq.18, we conclude that the numerical error of an DDIM sampling step is determined by four factors, namely, the step size $\Delta\phi = \phi_{t-1} - \phi_t$, the normalized model output $u_\theta^{(t)}/r_t$, the relative error of the data type $\delta$, and $\sin(\psi_t - \phi_t)$, which is decided by the parameterization of the diffusion model.

Notably, not all these four factors are useful to achieve the goal of minimizing the numerical error. For example, tuning down the step size only decreases the numerical error of each step. As a result, the total step number of DDIM sampling is increased proportionally, which cancels out the effect of error reduction of each single step. The factor $u_\theta^{(t)}/r_t$ is not adjustable since it constantly has a unit standard deviation. This can be verified by the following calculation:

$$\mathbb{E}[(u^{(t)}/r_t)^2] = \mathbb{E}[(\cos\psi_t x + \sin\psi_t \epsilon)^2] = \cos^2\psi_t \mathbb{E}[x^2] + \sin^2\psi_t. \tag{19}$$

In common practice, image tokens $x$ are normalized into unit standard deviation. Therefore, $\mathbb{E}[(u^{(t)}/r_t)^2] = \cos^2\psi_t + \sin^2\psi_t = 1$.

If we decide to scale up our model, it is better to leverage the pre-trained LLMs as well as the highly efficient training infrastructure that is specifically optimized for LLMs. This makes `bfloat16` almost the only choice. As a result, the relative error $\delta$ is fixed to $1/128$.

Now, our only choice is to adjust the diffusion model parameterization method, so that $|\sin(\psi_t - \phi_t)|$ is maximized. A simple solution is to set $\psi_t - \phi_t = \pi/2$, resulting in the following parameterization:

$$u^{(t)} = r_t \cos(\phi_t + \pi/2)x + r_t \sin(\phi_t + \pi/2)\epsilon = r_t(\cos\phi_t \epsilon - \sin\phi_t x). \tag{20}$$

Note that $r_t$ is still undetermined, which reflects the scale of $u^{(t)}$. From the analysis above, $r_t$ does not affect the numerical error term, since it is canceled out by the normalization of the model output, as seen in the factor $u_\theta^{(t)}/r_t$. Therefore, $r_t$ can be chosen freely, or based on other considerations. We consider that the smooth optimization of a neural network often requires the activation and output not too large or small. Therefore, we require a unit standard deviation for $u^{(t)}$, making $r_t = 1$ constantly.

The final parameterization of our diffusion model is as follows:

$$u^{(t)} = \cos\phi_t \epsilon - \sin\phi_t x. \tag{21}$$

We notice that this parameterization is coincidentally the "v-prediction" parameterization(Salimans & Ho, 2022). Note that, however, "v-prediction" is initially proposed for the efficient distillation of diffusion models, rather than reducing the numerical error of diffusion models. To the best of our knowledge, our work is the first to derive "v-prediction" parameterization from the first principle of minimizing the numerical error in diffusion models.

## A.3 DERIVING THEORETICAL NUMERICAL ERROR FOR n-PREDICTION MODELS

The n-prediction parameterization corresponds to $\psi_t = \frac{\pi}{2}$ in the angular parameterization form given by eq.11. Substituting $\psi_t = \frac{\pi}{2}$ and $u_\theta^{(t)}/r_t = \epsilon_\theta$ into eq.18, we get:

$$\tilde{x}^{(t-1)} = x^{(t-1)} + \sin(\phi_{t-1} - \phi_t)\frac{\epsilon_\theta}{\cos(\phi_t)}\delta. \tag{22}$$

Further, we cancel out the step size factor $\sin(\phi_{t-1} - \phi_t)$ within the numerical error term, only focusing on "the numerical error introduced per **unit** DDIM step":

$$e^{(t)} = \frac{\epsilon_\theta}{\cos\phi_t}\delta. \tag{23}$$

Next, we will show that $e^{(t)}$ can also be interpreted as the equivalent v-prediction numerical error for an n-prediction model.

For an n-prediction model, $u_\theta^{(t)} = \epsilon_\theta$. In order to calculate the equivalent $v_\theta^{(t)}$ value, we need to represent $v_\theta^{(t)}$ with the predicted $\epsilon_\theta$ and the known $x^{(t)}$, which is calculated as follows:

$$v_\theta^{(t)} = \cos\phi_t \epsilon_\theta - \sin\phi_t \hat{x}_\theta(x^{(t)}) = \cos\phi_t \epsilon_\theta - \sin\phi_t \frac{x^{(t)} - \sin\phi_t \epsilon_\theta}{\cos\phi_t} = \frac{\epsilon_\theta}{\cos\phi_t} - \tan\phi_t x^{(t)}. \tag{24}$$

Considering the numerical error, we get:

$$\tilde{v}_\theta^{(t)} = \frac{\epsilon_\theta(1+\delta)}{\cos\phi_t} - \tan\phi_t x^{(t)} = v_\theta^{(t)} + \frac{\epsilon_\theta}{\cos\phi_t}\delta. \tag{25}$$

Note that the numerical error term in the above equation is exactly $e^{(t)}$, proving that $e^{(t)}$ can be interpreted as the equivalent v-prediction numerical error for an n-prediction model.

Taking numerical error effect into the v-prediction-based diffusion loss, we get:

$$\mathbb{E}[(v^{(t)} - \tilde{v}_\theta^{(t)})^2] = \mathbb{E}[(v^{(t)} - v_\theta^{(t)} - e^{(t)})^2] = \mathbb{E}[(v^{(t)} - v_\theta^{(t)})^2] - 2\mathbb{E}[(v^{(t)} - v_\theta^{(t)})e^{(t)}] + \mathbb{E}[(e^{(t)})^2]. \tag{26}$$

Due to the fact that numerical error $e^{(t)}$ is independent from the training loss and that the expectation of $e^{(t)}$ is 0, we get $\mathbb{E}[(v^{(t)} - v_\theta^{(t)})e^{(t)}] = 0$. Therefore, the only numerical error term is $\mathbb{E}[(e^{(t)})^2]$. Given that the standard deviation of $\epsilon_\theta$ is 1, and considering that we use `bfloat16` as training data type, which means $\delta = 1/128$, we get

$$\mathbb{E}[(e^{(t)})^2] = \frac{1}{(128\cos(\phi_t))^2} = \frac{1}{128^2\bar{\alpha}_t}. \tag{27}$$

This is the theoretical numerical error of the v-prediction diffusion loss for an n-prediction model.

### A.4 CFG WITH $v$-PREDICTION

From Equation $v_i^{(t)} = \sqrt{\bar{\alpha}_t}\epsilon - \sqrt{1-\bar{\alpha}_t}x_i$, we can derive the following equation.

$$\epsilon = \sqrt{1-\bar{\alpha}^{(t)}}x^{(t)} + \sqrt{\bar{\alpha}^{(t)}}v \tag{28}$$

For the CFG of $\epsilon$, it can be simplified as follows.

$$\begin{aligned}
\epsilon &= \epsilon_u + \omega(\epsilon_c - \epsilon_u) \\
&= \sqrt{1-\bar{\alpha}^{(t)}}x^{(t)} + \sqrt{\bar{\alpha}^{(t)}}v_u + \omega\sqrt{\bar{\alpha}^{(t)}}(v_c - v_u) \\
&= \sqrt{1-\bar{\alpha}^{(t)}}x^{(t)} + \sqrt{\bar{\alpha}^{(t)}}(v_u + \omega(v_c - v_u))
\end{aligned} \tag{29}$$

Ultimately, we obtain $v = v_u + \omega(v_c - v_u)$. The CFG of $v$ and $\epsilon$ are equivalent.

### A.5 EXAMPLES: THE EFFECT OF THE STAGE 2 TRAINING

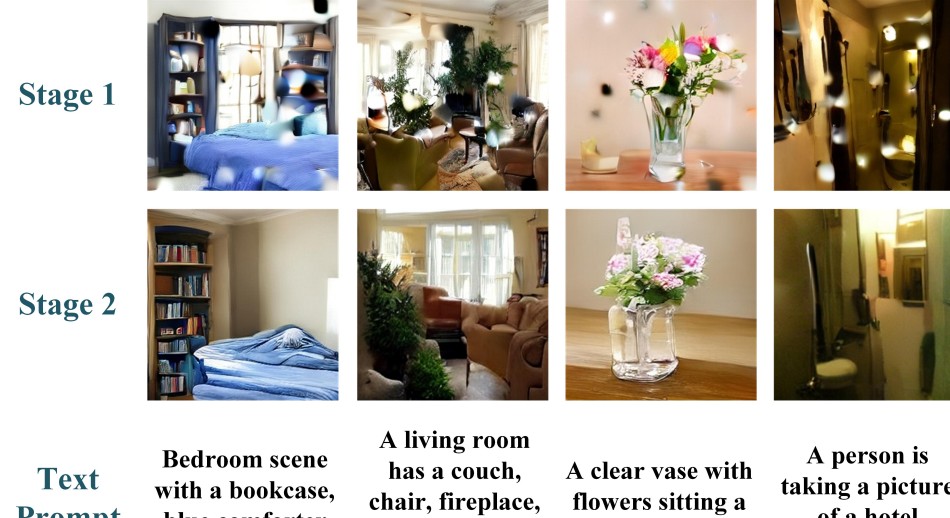

Figure 5: The impact of the second training stage on image generation capability.

### A.6 EXAMPLES: IMAGE GENERATION

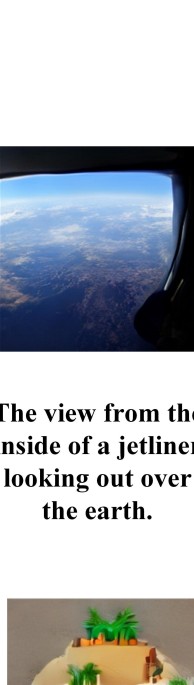 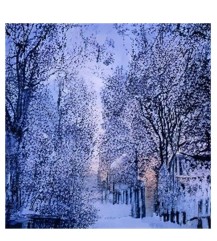 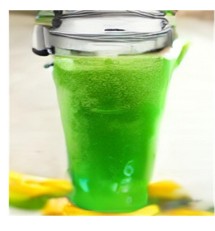 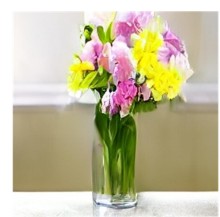

| **The view from the inside of a jetliner looking out over the earth.** | **Artistic photograph of architecture and frost-laden trees at dawn** | **A blender full of a green colored smoothie** | **Flowers are in a vase of water on a tabletop.** |

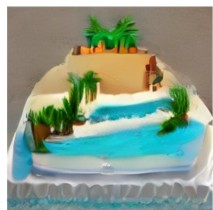 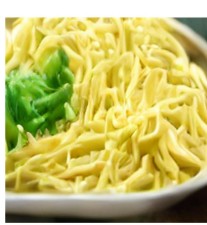 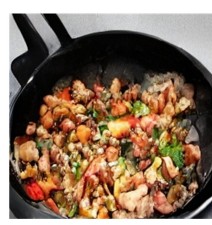 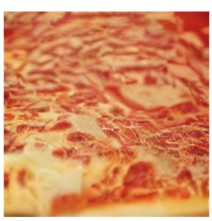

| **A birthday cake fashioned to look like a beach with surfboard and palm trees.** | **A close-up of a plate of pasta containing broccoli** | **A skillet contains diced meat and mixed vegetables.** | **a close up of a sheet of pizza on a table** |

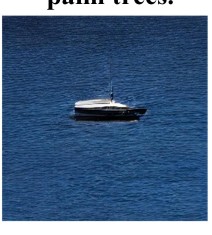 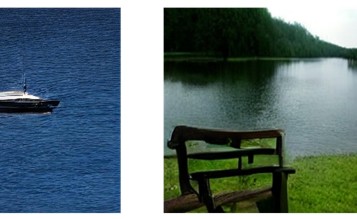 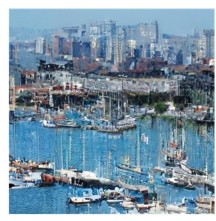 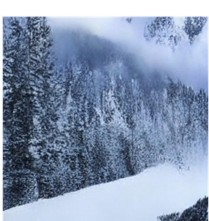

| **A boat sailing on top of a body of water.** | **A park bench on the side of a lake.** | **A harbor with many boats and city in the background.** | **A snow covered forest on the side of a mountain.** |

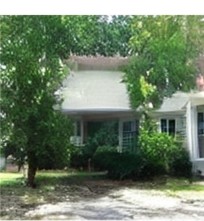 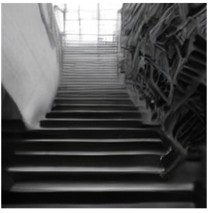 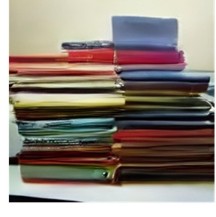 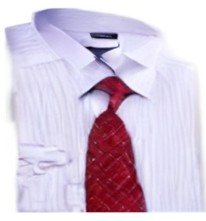

| **The front of a house is next to some trees.** | **Photo taking in a building looking at a staircase.** | **Several books are stacked on a table.** | **Not really a good choice for this shirt and tie combination.** |

Figure 6: Generated images from MMAR-7B