# OpenReview forum: "MMAR: Towards Lossless Multi-Modal Auto-Regressive Prababilistic Modeling"
_ICLR.cc/2025/Conference — ICLR 2025 Conference Withdrawn Submission_

### Official Review · Reviewer_zEBq · 2024-10-29

**Soundness:** 2
**Presentation:** 1
**Contribution:** 2
**Rating:** 3
**Confidence:** 4

**Summary:**

Previous work focusing on a unified image generation and understanding framework faces several challenges, such as information loss caused by vector quantization and the negative impact of Gaussian noise on image understanding performance. This paper effectively addresses these issues by presenting a novel lossless multi-modal model named MMAR. By introducing strategies like v-prediction and two-stage training, MMAR demonstrates excellent performance in both image understanding and generation.

**Strengths:**

1. Introducing MAR into image understanding is an innovative idea that effectively addresses the issues exposed in previous work.
2. This paper presents the instability in training caused by low-precision training and demonstrates that v-prediction is an effective solution. I believe this finding will certainly inspire future work.

**Weaknesses:**

1. **The citation format of this paper is a disaster.** Many previous works are not cited when mentioned for the first time (e.g., MARS, LlamaGen, and Chamelon in Line 53, VQ-VAE in Line 76, Transfusion in Line 80, ROPE in Line 335, SiLU in Line 340, and Imagegen in Table 2); abuse of \citet and \citep (e.g., Line 109, Line141-142, Line 160-161, and Line 428), no space between the text and the reference (e.g., Line 144-146, Line 157-158, Line 173, Line 182, Line 184, and Line 187). Because there are so many wrong formats, I do not list them individually.
2. **The article's logic needs to be improved.** In section 3.3.1, when *EmbeddingViT* and *VisProjector* are first introduced, no explanation or citation is provided, which makes me quite confused. When introducing new terms, the authors should clearly indicate where their definitions can be found rather than making readers spend time searching throughout the entire paper. Additionally, the abbreviation *CFG* is used extensively without any introduction, and the authors inconsistently use both *CFG* and *cfg* (line 351). Even in Section 3.3.4, which discusses Classifier-Free Guidance, I could not find an explanation clarifying that *CFG* refers to Classifier-Free Guidance.
3. **The assumption in Section 3.2 is unreasonable.**  In Section 3.2, the authors assume that $\epsilon_{\theta}(x^t, t, z_i)$ follows a standard normal distribution without providing any justification for the validity of this assumption. In fact, this assumption is unreasonable. Based on [1], the noise prediction network $\epsilon_{\theta}(x^t, t, z_i)$ models $E[\epsilon|x^t]$. As $t$ approx $0$, $\epsilon_{\theta}(x^t, t, z_i) \approx E[\epsilon|x^t]=E[\epsilon]=0$. The approximation error arises from the disparity between the model distribution and the true data distribution.
4. **Notations are confusing.** In Section 3.1, for autoregressive modeling, the authors use the same notation for both text and images (i.e.,  $p_{\theta}(x_i|x_{<i})$). This consistent notation suggests that both text and images employ a left-to-right next-token prediction modeling approach. However, in the experimental section, the modeling for images utilizes different masked autoregressive models.
5. **The authors claim that this paper model the joint probabilistic $P(T, I)$. However, $P(T, I)$ is never defined.** In Line 225, the authors claim that Eq. (4) is the overall loss for the image-text joint probability. Is Eq. (4) an exact likelihood for $p(T, I)$, or is it an upper bound? I do not see any convincing proof explaining the relationship between Eq. (4) and $p(T, I)$. In fact, according to the authors' notation, Eq. (4) does not reflect any interaction between text and images; it appears to model the marginal distributions of text and images separately.
6. **The authors claim that this paper presents scaling-up laws of MMAR in the abstract and introduction. However, the main text does not present any scaling-up laws.** The scaling laws [2,3] are established through a series of experiments that identify quantitative power-law relationships between loss or other metrics and factors such as model size, data size, and training compute. In this paper, all experiments related to scaling cannot be referred to as laws; "scaling behavior" is a more appropriate term.
7. **The authors claim that v-prediction is the 'only' way to reduce the error between data distribution and model distribution in Line 113 and Line 258, such a claim is highly questionable.**  Possible solutions could also include adjusting optimizer parameters, using more advanced network architectures, and exploring different noise schedules. However, the authors hastily claim that v-prediction is the only solution without attempting these alternatives.
8. **The evaluation of text-to-image results is insufficient.** More metrics, such as Geneval[4] and T2i-bench[5], should be considered. Only zero-shot FID results are insufficient.

I strongly support the authors' motivation, but this paper is rife with flawed logic, formatting errors, unreasonable assumptions, confusing notations, overclaims, and insufficient evaluation. These factors lead me to reject this paper.


[1] Bao et al. Analytic-dpm: an analytic estimate of the optimal reverse variance in diffusion probabilistic models. ICLR2022.

[2] Kaplan et al. Scaling laws for neural language models. arXiv.

[3] Hoffmann et al. Training compute-optimal large language models. arXiv.

[4] Ghosh et al. Geneval: An object-focused framework for evaluating text-to-image alignment. NeurIPS 2023

[5] Huang et al.  T2i-compbench: A comprehensive benchmark for open-world compositional text-to-image generation. ICCV 2023

**Questions:**

1. The authors claim that they evaluate MMAR on 18 image understanding benchmarks. However, they only present results for 5 benchmarks in Table 1 and subsequently report the average score across all 18 benchmarks. Where are the results for the remaining 13 benchmarks? I believe that the authors should present these results, even if in an appendix. I couldn't find them in the main text or the appendix. Do I overlook something?

---

### Official Review · Reviewer_iUjY · 2024-10-31

**Soundness:** 2
**Presentation:** 3
**Contribution:** 2
**Rating:** 5
**Confidence:** 3

**Summary:**

This paper proposes a multi-model autoregressive joint image-text modeling framework. The main three contributions are modeling continuous image presentations on top of LLM to eliminate the information loss associated with VQ operations, deriving the optimal diffusion parameterization for low-precision training, and introducing a two-stage training strategy with large CFG scale to improve the model's image generation and understanding capability.

**Strengths:**

- The proposed model complements existing MLLM works as it is a continuous space auto-regressive model with the image prior P(I) and text probability conditioned on image P(T|I) as shown in Figure 1.
- Low-precision training of diffusion models is important for efficient deployment on resource-limited edge devices.

**Weaknesses:**

- Lack of novelty. The first contribution does not look so novel as it was the main contribution of MAR[1]. The deduction of low-precision training of diffusion models, i.e., Eq. (5) & (6), looks straightforward to me while I do not think it could be a main contribution.
- The proposed two-stage training strategy is not proven to outperform existing state-of-the-art MLLM works according to the experiments on visual understanding and visual generation. Generally, the MMAR just performs the second best when compared to other joint image-text probabilistic models.




[1] Li, Tianhong, Yonglong Tian, He Li, Mingyang Deng, and Kaiming He. "Autoregressive Image Generation without Vector Quantization." arXiv preprint arXiv:2406.11838 (2024).

**Questions:**

Typos (do not affect my rating):
1. Title: "PRABABILISTIC" should be "PROBABILISTIC".
2. Section 2.1, "... more and more research has be focused on ..." should be "has been focused on".

---

### Official Review · Reviewer_ezsF · 2024-11-03

**Soundness:** 2
**Presentation:** 3
**Contribution:** 3
**Rating:** 5
**Confidence:** 4

**Summary:**

The paper proposes to combine the MAR image generation approach, which uses sequential mask modeling a la MaskGIT and per-token latent diffusion, with autoregressive text modeling. The goal of this combination is the joint modeling of images and text, enabling image understanding applications (image-to-text, I2T) and image generation (text-to-image, T2I) with a single unified model. The proposed MMAR models are built on top pretrained QWen LLMs and a LDM KL-16 VAE. The authors also explore different diffusion formulations and analyze the effect of low-bit training on the robustness of different formulations. MMAR is evaluated on a broad suite of multimodal understanding tasks, showing good performance to a set of established models from the literature. Image generation quality is evaluated based on COCO-FID30k.

**Strengths:**

- MAR has not been previously explored in the context of multimodal joint generation and understanding, so this application is new.
- The paper presents some interesting ablations (v-parametrization vs n-parametrization, comparison with VQ-based variant).
- The proposed method obtains quite strong understanding performance compared to a set of strong baselines from the literature (such as LLaVA, Chameleon, DreamLMM)

**Weaknesses:**

- I’m not convinced by the argument in the introduction that the proposed MMAR, which uses masked modeling, is fundamentally better suited for image understanding tasks than Transfusion, which uses diffusion on the full image. Specifically, the authors argue that full image diffusion hurts understanding because the model is trained on noisy images (with varying levels of noise). However, MMAR is also trained on noisy images when modeling P(image|text), just instead of additive noise, the noise comes from dropped tokens. Furthermore, when modeling P(text|image) both Transfusion and MMAR can operate on noiseless and fully unmasked, respectively.
- The text-to-image quality of MMAR is somewhat weak compared to the image understanding quality. It would be good to see why this is, e.g. because the model relies on a pretrained LLM which is a adapted with LoRA. A image generation-only baseline, potentially trained from scratch, could elucidate this question.
- While the error/stability analysis of the n vs. v-parametrization makes sense to me, the authors never experimentally show that going from high-precision training to low precision training actually leads to issues. Would it be possible to show some training curves?

**Questions:**

- The Limitations section mentions that the proposed approach is slow. Why is it slow, since speed was one of the selling points of MAR? Also what role does the MAR decoder play, which predicts the mask location, which MMAR does not use? Could this decoder speed up inference?
- In the description of the Diffusion MLP (L339 and following) the authors say they condition the MLP performing per-patch diffusion. What is this MLP conditioned on?
- Did the authors run an ablation to inform the design choice to not use a positional encoding for the image tokens? Intuitively they should be quite useful for masked modeling, where tokens are generated in variable chunks. On a related note, some details on the image sampling (such as the masking schedule) seem to be missing.
- Could you formally derive the limit in L253-256?

---

### Official Review · Reviewer_QGvQ · 2024-11-04

**Soundness:** 2
**Presentation:** 3
**Contribution:** 2
**Rating:** 3
**Confidence:** 4

**Summary:**

The paper proposes an image-text joint probabilistic modeling method called MultiModal AutoRegressive (MMAR). The authors address the limitations that existing methods using discrete image tokenization could lead to information loss. MMAR utilizes continuous image representation to avoid these drawbacks by employing a diffusion-based continuous probabilistic sampler integrated with large language models (LLMs).

**Strengths:**

1. The analysis of using v-prediction from the perspective of numerical error is interesting and useful.

2. Exploring the paradigm of "next-token-prediction" for large scale multimodal generation is an important topic.

**Weaknesses:**

1. The paper is a simple extension of MAR and is of insufficient novelty.

2. I am not convinced by the paper's claim that discretization introduces a bottleneck and that using continuous probabilistic models is necessary for image modeling in multimodal modeling due to the following reasons.

i) A unified representation and network structure for multimodal modeling is simpler and easy to scale.

ii) It is hard to claim introducing continuous image representation can outperform purely discrete representations. It seems that purely discrete representations have not yet reached their full potential. For example, EMU3 [1] uses purely discrete representations and achieves state-of-the-art performance in many areas.

iii) Data is inherently discrete in computers.

3. Current experimental results in the paper are not strong enough to support the main claim. For example, does the MMAR beat domain-specific generative model such as stable diffusion like EMU3? The generation results in the appendix seems to be of low quality.

[1] Wang, Xinlong, et al. "Emu3: Next-token prediction is all you need." arXiv preprint arXiv:2409.18869 (2024).

**Questions:**

EDM [2] also derive a preconditioned technique for improving training like “v-prediction”. Can the author provide an analysis on the numerical error when using EDM precondition?

[2] Karras, Tero, et al. "Elucidating the design space of diffusion-based generative models." Advances in neural information processing systems 35 (2022): 26565-26577.

---

### Note · Authors · 2024-11-14

**Comment:**

We are very grateful for the constructive suggestions and the effort expended by all the reviewers.

**Withdrawal Confirmation:**

I have read and agree with the venue's withdrawal policy on behalf of myself and my co-authors.